# Cathelicidin Attenuates Hyperoxia-Induced Lung Injury by Inhibiting Ferroptosis in Newborn Rats

**DOI:** 10.3390/antiox11122405

**Published:** 2022-12-04

**Authors:** Hsiu-Chu Chou, Chung-Ming Chen

**Affiliations:** 1Department of Anatomy and Cell Biology, School of Medicine, College of Medicine, Taipei Medical University, Taipei 110, Taiwan; 2Department of Pediatrics, Taipei Medical University Hospital, Taipei 110, Taiwan; 3Department of Pediatrics, School of Medicine, College of Medicine, Taipei Medical University, Taipei 110, Taiwan

**Keywords:** cathelicidin, hyperoxia, newborn, ferroptosis, radial alveolar count

## Abstract

High oxygen concentrations are often required to treat newborn infants with respiratory distress but have adverse effects, such as increased oxidative stress and ferroptosis and impaired alveolarization. Cathelicidins are a family of antimicrobial peptides that exhibit antioxidant activity, and they can reduce hyperoxia-induced oxidative stress. This study evaluated the effects of cathelicidin treatment on lung ferroptosis and alveolarization in hyperoxia-exposed newborn rats. Sprague Dawley rat pups were either reared in room air (RA) or hyperoxia (85% O_2_) and then randomly given cathelicidin (8 mg/kg) in 0.05 mL of normal saline (NS), or NS was administered intraperitoneally on postnatal days from 1–6. The four groups obtained were as follows: RA + NS, RA + cathelicidin, O_2_ + NS, and O_2_ + cathelicidin. On postnatal day 7, lungs were harvested for histological, biochemical, and Western blot analyses. The rats nurtured in hyperoxia and treated with NS exhibited significantly lower body weight and cathelicidin expression, higher Fe^2+^, malondialdehyde, iron deposition, mitochondrial damage (TOMM20), and interleukin-1β (IL-1β), and significantly lower glutathione, glutathione peroxidase 4, and radial alveolar count (RAC) compared to the rats kept in RA and treated with NS or cathelicidin. Cathelicidin treatment mitigated hyperoxia-induced lung injury, as demonstrated by higher RAC and lower TOMM20 and IL-1β levels. The attenuation of lung injury was accompanied by decreased ferroptosis. These findings indicated that cathelicidin mitigated hyperoxia-induced lung injury in the rats, most likely by inhibiting ferroptosis.

## 1. Introduction

Approximately 50% of preterm infants under 30 weeks of gestation suffer from respiratory distress syndrome, the most common cause of respiratory distress in preterm infants [1]. Infants with respiratory problems are frequently treated with supplemental oxygen [2]. However, the oxygen treatment given to infants has downsides. Preclinical studies have demonstrated that prolonged exposure of neonatal rodents to hyperoxia results in increased lung inflammation and decreased alveolarization and angiogenesis; this resembles bronchopulmonary dysplasia in humans [3,4]. Despite the use of appropriate ventilation techniques, early surfactant therapy, and noninvasive positive pressure ventilation, bronchopulmonary dysplasia remains a major cause of morbidity and mortality in preterm infants during the first year of life [5,6]. Existing therapies are not effective in preventing or treating hyperoxia-induced lung injury.

Hyperoxia increased lung oxidative stress in neonatal rats [7]. Oxidative stress is associated with complications in preterm infants, including preterm retinopathy, bronchopulmonary dysplasia, and necrotizing enterocolitis [8,9]. Oxidative stress induced mitochondrial dysfunction and ferroptosis in cell lines derived from a pheochromocytoma of the rat adrenal medulla [10]. Antibacterial, antiviral, and antifungal properties are found in the family of antimicrobial peptides known as cathelicidins, they perform multiple functions and improve innate immunity [11,12]. An iron-dependent form of non-apoptotic cell death known as ferroptosis is characterized by the accumulation of reactive oxygen species and lipid hydroperoxides in the cell membrane. [13]. We have previously shown that neonatal mice exposed to hyperoxia (85% O_2_) for 7 days experienced ferroptosis and impaired lung development, and that newborn rats exposed to hyperoxia on postnatal day 7 experienced lessened lung damage due to cathelicidin treatment [14,15]. This study evaluates the hypothesis that hyperoxia exposure affects lung development and that cathelicidin, by preventing ferroptosis in newborn rats, ameliorates hyperoxia-induced lung damage. Thus, the therapeutic effects of cathelicidin on ferroptosis biomarkers and lung development in neonatal rats exposed to hyperoxia were evaluated in this study.

## 2. Materials and Methods

### 2.1. Experimental Groups

We performed the experiments in accordance with the guidelines and regulations of the Institutional Animal Care and Use Committee of Taipei Medical University (LAC-2018-0342). Time-dated pregnant Sprague Dawley rats on gestational day 14, supplied by BioLASCO Taiwan Co., Ltd. (Taipei, Taiwan), were housed in individual cages, and were given free access to laboratory food and water on a 12:12-h light–dark cycle. The rats were allowed to deliver vaginally at the end of their pregnancies. Four female rats gave birth to forty pups. The litters were pooled, and ten pups were randomly redistributed among the new mothers within 12 h of delivery; the pups were then randomly chosen to be raised in room air (RA) or O_2_-enriched air. Between postnatal days 1 and 7, the group pups with hyperoxia (O_2_, normobaric) were raised in an atmosphere containing 85% O_2_. The RA group pups were raised in RA from postnatal day 1–7. Every 24 h, the nursing mothers were switched between the O_2_ treatment and RA control litters to prevent oxygen poisoning. A clear 40 × 50 × 60 cm^3^ plexiglass box that was kept in an O_2_-rich (85%) environment received a constant intake of O_2_ at a rate of 4 L/min. A ProOx P110 (NexBiOxy, Hsinchu, Taiwan) was used to track the O_2_ levels. On postnatal days between 1 and 6, cathelicidin (8 mg/kg; Bioworld Technology, Inc., St Louis Park, MN, USA) in 0.05 mL of normal saline (NS) was administered intraperitoneally to half of the pups in the RA and hyperoxia groups. Four experimental groups were established: (1) RA + NS, (2) RA + cathelicidin, (3) O_2_ + NS, and (4) O_2_ + cathelicidin. The rat pups were given 1% isoflurane (Halocarbon Laboratories, River Edge, NJ, USA) anesthesia on postnatal day 7, and their lungs were extracted for histological and biochemical examinations.

### 2.2. Measurement of Ferroptosis Indicators

Lipid peroxidation and iron accumulation are the two main causes of ferroptosis [16]. Therefore, we measured malondialdehyde (MDA), a lipid peroxidation end product [17], and glutathione peroxidase 4 (GPX4), a phospholipid hydroperoxidase that inhibits lipid peroxidation [18]. Glutathione (GSH) is an antioxidant, and its depletion induces lipid peroxidation by activating lipoxygenases and inhibiting GPX4 activity [19]. To eliminate cellular debris, the lung tissues were homogenized, sonicated, and centrifuged at 500× *g* for 20 min at 4 °C. As biomarkers of ferroptosis, the Fe^2+^ level, MDA level, GSH level, and GPX4 activity were measured in lung tissues using an iron assay kit (catalog no. E-BC-K139-M, Elabscience, Houston, TX, USA), MDA assay kit (catalog no. MBS2605193, MyBioSource, San Diego, CA, USA), GSH assay kit (catalog no. MBS267424, MyBioSource), and GPX4 enzyme-linked immunosorbent assay kit (catalog no. MBS934198, MyBioSource), respectively, according to the directions provided by the respective manufacturers.

### 2.3. Determination of Iron Deposition

We used an iron staining kit based on the Prussian blue reaction (ScyTek Laboratories Logan, UT, USA) to identify iron accumulation in lung tissues [14]. Briefly, following deparaffinization and rehydration, 5-µm lung sections were counterstained with nuclear rapid red solution, dehydrated with alcohol, cleaned with xylene, and mounted on cover plates for examination. Ferric iron is colored blue in this stain, while the nuclei are colored red. Iron deposition was quantified by counting the positive Prussian blue stained cells in randomly selected 50 high power fields (magnification 400×)/group, which were taken and uploaded to a computer screen for analysis using the Image Pro Plus software (Media Cybernetics, Silver Spring, MD, USA). For final calculations, data generated from all sampled sections, fields, and counts were used.

### 2.4. Western Blots of GPX4 and TOMM20

Lung tissues were homogenized in a cold buffer consisting of a mixture of protease inhibitors, 50 mmol/L Tris-HCl (pH 7.5), 1 mmol/L ethylene glycol tetraacetic acid, and 1 mmol/L ethylenediaminetetraacetic acid (entire mini tablets; Roche, Mannheim, Germany). A 12% sodium dodecyl sulfate-polyacrylamide gel electrophoresis was used to resolve proteins (30 g) under reduced conditions, and the proteins were electroblotted to a polyvinylidene difluoride membrane (Immobilon P, Millipore). After being blocked with 5% non-fat dry milk, membranes were incubated with anti-β-actin (1:1000; C4 sc-47778, Santa Cruz Biotechnology) and anti-GPX4 (1:750; SC-7269, Santa Cruz Biotechnology), followed by horseradish peroxidase-conjugated goat anti-mouse (Pierce Biotechnology, Rockford, IL, USA). The protein bands were identified with a Pierce BioSpectrum AC System. The intensity of the TOMM20 and β-actin bands on AIDA was measured using densitometric analysis. After normalizing with β-actin, the densitometry unit of TOMM20 protein expression in the RA + NS group was designated as 1.

### 2.5. Immunohistochemistry of TOMM20 and Cathelicidin

On 5-μm paraffin sections, immunoperoxidase visualization was used to conduct immunohistochemical staining. After routine deparaffinization, the slides were submerged in 0.01 M sodium citrate buffer (pH 6.0) for heat-induced epitope retrieval. The sections were preincubated in 0.1 M PBS containing 10% normal goat serum and 0.3% H_2_O_2_ for 1 h at room temperature in order to inhibit endogenous peroxidase activity and nonspecific antibody binding. After that, the sections were incubated with the primary antibody, recombinant anti-TOMM20 antibody (1:100; Abcam, Cambridge, MA, USA) and anti-cathelicidin antibody (1:50; Affinity Biosciences, Cincinnati, OH, USA) for 20 h at 4 degrees. The sections were subsequently treated with biotinylated goat anti-rabbit IgG for 1 h at 37 °C (1:200, Jackson ImmunoResesarch Laboratories Inc., PA, USA). After reactions using reagents from the Avidin-Biotin Complex Kit (Vector Laboratories, Burlingame, CA, USA), reaction products were visualized using the Diaminobenzidine Substrate Kit (Vector Laboratories, Inc.) according to the manufacturer’s recommendations. All immunostained sections were examined and photographed with a Nikon Eclipse E600 Microscope (Nikon, Tokyo, Japan). Five randomly selected fields from each section at 400× magnification were photographed using a digital camera and imported into a computerized image analysis system (Image-Pro Plus 6.0 for Windows, Media Cybernetics, Silver Spring, MD, USA). Immunoreactive positive cathelicidin cells were quantified using an automated object counting and measurement process, yielding a percentage of positively stained cells; these values were expressed as a labeling index (%). The histological quantification was performed in a single-blind fashion by a pathologist.

### 2.6. Lung Histology

For 24 h, the lung tissue was submerged in 0.1 M phosphate buffer (pH 7.4) containing 4% paraformaldehyde at 4 °C. The tissues were subsequently dehydrated using alcohol, washed with xylene, and embedded in paraffin. Additionally, lung morphometry was used to evaluate 5-μm sections after they had been stained with hematoxylin and eosin and inspected under a light microscope. The Cooney et al. [20] approach was modified to quantify the radial alveolar count (RAC) in order to assess the structural development of alveoli. The average number of alveoli transected by a perpendicular line drawn from the center of a respiratory bronchiole to the nearest septal division is referred to as the RAC. RAC was measured by counting the number of alveoli passed by a vertical line from the center of the respiratory bronchioles to the border of the pleura (Appendix A). We identified one RAC measure as one unit and detected from 90–100 units (magnification 200×)/group. Data generated from all sampled sections, fields, and counts were used for final calculations.

### 2.7. Lung Cytokine Assay

To remove cellular debris, 100 mg of lung tissue from each pup was homogenized, sonicated, and centrifuged at 500× *g* for 20 min at 4 °C as directed by the manufacturer. A kit for enzyme-linked immunosorbent assay (ELISA) (R&D systems, Abingdon, UK) was used to measure the amounts of interleukin-1β (IL-1β) in the supernatants.

### 2.8. Statistical Analysis

A mean ± standard deviation is used to express the data. One-way analysis of variance and Tukey’s post hoc test for multiple group comparisons were both used in the statistical analyses. *p* values of < 0.05 were used to determine statistical significance.

## 3. Results

### 3.1. Survival Rate

Four female rats delivered forty pups. Ten pups were redistributed at random among the new mothers after the litters were pooled. All ten rats housed in RA and treated with NS or cathelicidin survived, as did ten rats housed in hyperoxia and treated with cathelicidin. Pups housed in hyperoxia and treated with NS died on postnatal day 6.

### 3.2. Cathelicidin Improved Hyperoxia-Induced Decrease in Body Weight

Birth and postnatal day 1 body weights were similar in the RA and hyperoxia groups. (Figure 1). From day 2 to day 7 of life, hyperoxia-fed and NS-treated rats weighed significantly less than RA-fed and NS- or cathelicidin-treated rats. The rats treated with cathelicidin exhibited greater body weight compared to those reared in hyperoxia and treated with NS from postnatal day 5, and the differences reached statistical significance on postnatal days 6 and 7.

### 3.3. Hyperoxia Decreased Lung Cathelicidin Expression

Cathelicidin immunoreactivities were mainly detected in airway epithelial cells (Figure 2A). The immunoreactivity significantly decreased in the rats exposed to hyperoxygenation and treated with NS when compared with rats reared in RA and treated with NS or cathelicidin and rats exposed to hyperoxygenation and treated with cathelicidin (Figure 2B).

### 3.4. Cathelicidin Improved Hyperoxia-Induced Ferroptosis

Compared with rats exposed to hyperoxygenation and treated with NS, rats raised in RA and treated with NS or cathelicidin had significantly lower levels of Fe and MDA and significantly higher levels of GSH and GPX4. (Figure 3A–E). The therapy with cathelicidin effectively restored the hyperoxia-induced alterations in ferroptosis biomarkers.

### 3.5. Cathelicidin Decreased Hyperoxia-Induced Increase in Iron Deposition

Prussian blue-stained representative lung sections are shown in Figure 4A. The majority of Prussian blue staining was found in the cytoplasm of type II alveolar cells and macrophages. The rats raised in RA and treated with NS or cathelicidin did not show any iron deposition that could be detected. In comparison to rats raised in RA and treated with cathelicidin, which significantly reduced the hyperoxia-induced increase in iron deposition, the rats raised in hyperoxia and treated with NS showed higher iron deposition and a significantly higher number of Prussian blue stained cells per high-power field (Figure 4B).

### 3.6. Cathelicidin Decreased Hyperoxia-Induced Increase in TOMM20

Using primary antibodies to recombinant TOMM20, a marker of mitochondrial outer membrane protein expression, we probed lung tissues and homogenates. Rats raised in RA and treated with cathelicidin significantly inhibited the hyperoxia-induced rise in TOMM20 expression, whereas those raised in hyperoxia and treated with NS showed increased TOMM20 expression (Figure 5A,B).

### 3.7. Cathelicidin Decreased Lung Cytokine and Improved Alveolarization

The IL-1β levels of rats reared in hyperoxia and treated with NS were significantly higher compared to those of rats reared in RA and treated with NS or cathelicidin (Figure 6). The rise in IL-1β caused by hyperoxia was significantly decreased by cathelicidin treatment. Representative lung slices stained with hematoxylin and eosin are shown in Figure 7A. Normal lung morphology and similar RAC were visible in the rats raised in RA and treated with NS or cathelicidin. Large thin-walled air spaces and much lower RAC were present in the rats raised in hyperoxia and treated with NS compared to those raised in RA and treated with NS or cathelicidin (Figure 7B). The decrease in RAC caused by hyperoxia was significantly increased by cathelicidin treatment.

## 4. Discussion

Our animal model revealed that exposure to increased oxygen during the first seven days following delivery can lead to reduced body weights, induced ferroptosis, impaired lung alveolarization, and elevated lung cytokine levels in rats. Cathelicidin treatment alleviated lung damage, as demonstrated by higher RAC and lower lung cytokine levels in rats treated with cathelicidin compared to rats treated with NS. The improvement of lung injury was accompanied by the normalization of ferroptosis biomarkers. The primary discovery is that cathelicidin may suppress ferroptosis, protecting against hyperoxia-induced lung injury, suggesting that cathelicidin may be a viable therapy for hyperoxia-induced lung injury.

It is frequently necessary to treat neonates with respiratory problems using supraphysiological oxygen. Due to the fact that term newborn rats are born at the saccular stage, which is roughly similar to the human gestational age from 26 to 28 weeks, term-born rat models are suitable for investigating the impact of hyperoxia on preterm neonates with respiratory distress [21]. Consequently, newborn rats serve as useful models for assessing lung injury and lung development.

Mean body weights at birth and after 1-day hyperoxia exposure on the first day of life was similar between the RA and hyperoxic groups. Hyperoxia exposure decreased the mean body weight from postnatal day 2 onward. Cathelicidin treatment increased the body weights of hyperoxia-exposed rats from postnatal days 5 to 7. RA-reared rats treated with NS or cathelicidin exhibited parallel body weight curves throughout the study period. These results showed that cathelicidin has no harmful results on the body weight of normal newborn rats and induces beneficial effects on body weight gain in hyperoxia-exposed newborn rats.

In mice, cathelicidin reduces the fatal effects of sepsis and endotoxin shock by inhibiting the lipopolysaccharide-induced apoptosis of endothelial cells [22]. The effects of cathelicidin on lung ferroptosis in hyperoxia-exposed newborn rats have not yet been elucidated. In this study, we demonstrated that cathelicidin reduced ferroptosis, as evidenced by decreased Fe2+, MDA, and iron deposition; in addition, it also improved GSH levels and GPX4 activity and attenuated hyperoxia-induced lung injury in neonatal rats. These findings suggest that cathelicidin is involved in regulating ferroptosis after hyperoxia-induced lung injury.

The consequence of hyperoxia on iron deposition in other organs was mostly unknown. Ferroptosis has been linked to a variety of system diseases, including nervous system diseases, heart disease, liver disease, gastrointestinal disease, lung disease, kidney disease, and others [23]. Almost all organs and tissues have been shown to be targets of oxygen toxicity, though some have received less attention [24]. Iron deposition was discovered in other organs in theory. More research is needed to investigate the effects of hyperoxia on iron deposition in other organs.

Hyperoxia exposure induces ferroptosis and increases oxidative stress and inflammation in neonatal rodent lungs [14,15]. In addition to being involved in hyperoxia-induced lung injury, ferroptosis promotes inflammation [25,26] and drives a local auto-amplification loop that leads to accelerated cell death and inflammation [27]. In the present study, cathelicidin treatment inhibited ferroptosis and attenuated hyperoxia-induced lung injury and inflammation. These findings indicate that cathelicidin targets initial ferroptosis and, thereby, prevents the development of the loop.

Ferroptosis, apoptosis, autophagy, and necroptosis can be distinguished from one another by having damaged mitochondria, small mitochondria with increased mitochondrial membrane density, diminished mitochondrial crista, and ruptured outer mitochondrial membrane [13]. In this study, mitochondrial damage was determined by detecting the mitochondrial outer membrane protein TOMM20, and exposure to hyperoxia enhanced TOMM20 expression. These findings are comparable with the study results of Beyer et al. [28], which showed exposure to hyperoxia for 48 h raised the levels of the mitochondrial outer membrane protein and lowered mitochondrial activity in the lungs of adult rats. The cathelicidin treatment diminished the hyperoxia-induced increase in TOMM20 expression. Cathelicidin decreased ferroptosis by enhancing mitochondrial activity, as demonstrated by these data.

It is less clear whether hyperoxia alters mitochondria in cells. Neonatal mice subjected to hyperoxia exhibit mitochondrial injury with complex-I malfunction, reduced ATP levels, and cell death in the lungs, according to Ratner et al. [29]. Kandasamy et al. found that hyperoxia-induced lung development stopping in newborn mice was associated with a decrease in mitochondrial activity and mitochondrial DNA variation [30]. Because cell reproduction and differentiation require energy, mitochondrial failure is most likely to blame for lung growth stopping, which affects both the alveolar and vascular compartments.

This research has some limitations. First, the cellular origin of IL-1β expression in the lung tissues of newborn rats exposed to hyperoxia was not investigated, though Piedboeuf et al. found no IL-1β expression in the lungs of adult male mice exposed to room air. After 3 and 4 days of hyperoxia exposure, IL-1β transcripts were found in pulmonary interstitial macrophages and neutrophils and were widespread in lung tissues [31]. Second, we did not concentrate on one or two pathways and investigate the entire signaling cascade rather than demonstrating several parallel mechanisms leading to protective effects. Third, other mitochondrial dysfunction markers, such as oxidative stress and reduced mitochondrial biogenesis, were not assessed, though oxidative stress and TOMM20 have been reported to be implicated in hyperoxia-induced lung injury [15,28].

In conclusion, as evidenced by the increase in body weight and RAC as well as the decrease in lung cytokine, cathelicidin attenuated hyperoxia-induced lung injury. The results also demonstrate that cathelicidin has no negative effects on normal lung development in neonates. Cathelicidin’s beneficial effects on hyperoxia-induced lung injury are associated with a reduction in ferroptosis. At present, there is no scientifically proven treatment for stopping lung damage brought on by hyperoxia exposure. Cathelicidin therapy may represent a novel approach to prevent hyperoxia-induced lung injury.

## Figures and Tables

**Figure 1 antioxidants-11-02405-f001:**
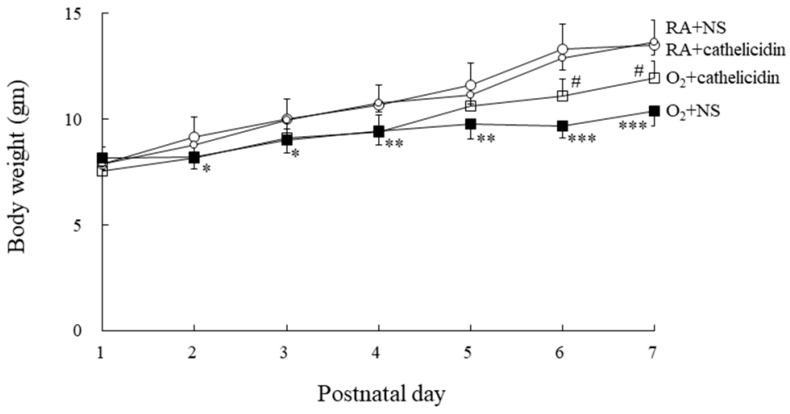
Body weights of the RA- or hyperoxia-reared rats treated with NS or cathelicidin from postnatal days 1 to 7. *n* = 9–10 rats in each group. * *p* < 0.05, ** *p* < 0.01, and *** *p* < 0.001 vs. RA + NS and RA + cathelicidin groups. # *p* < 0.01 vs. O_2_ + NS group.

**Figure 2 antioxidants-11-02405-f002:**
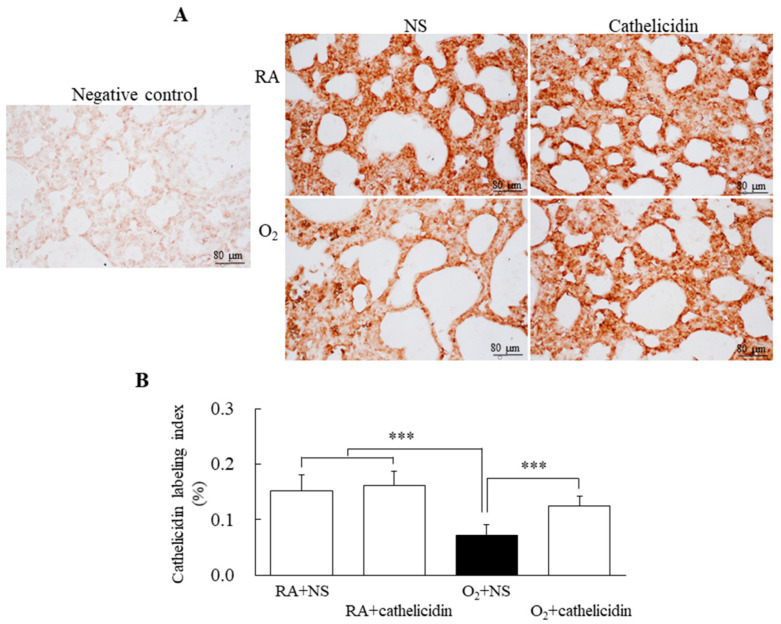
(**A**) Representative photomicrographs of immunohistochemistry of cathelicidin staining and (**B**) semi-quantitative analysis of cathelicidin immunoreactivity in 7-day-old rats exposed to RA or hyperoxia and treated with NS or cathelicidin. Treatment with cathelicidin significantly increased the hyperoxia-induced decrease in cathelicidin immunoreactivity. *n* = 9–10 rats in each group. *** *p* < 0.001.

**Figure 3 antioxidants-11-02405-f003:**
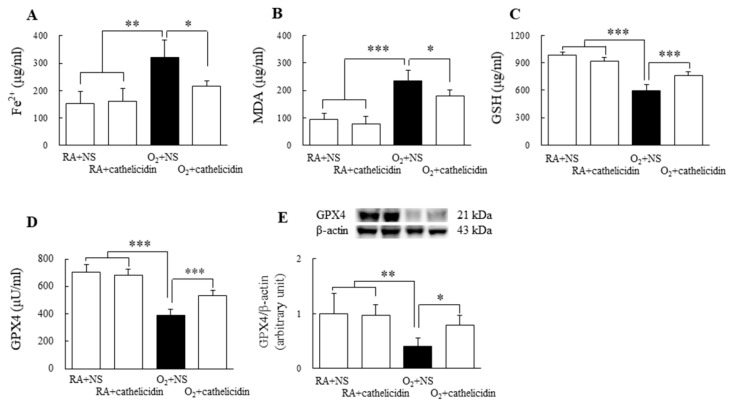
Ferroptosis biomarkers in 7-day-old rats exposed to RA or hyperoxia and treated with NS or cathelicidin. (**A**) Fe^2+^ level, (**B**) MDA level, (**C**) GSH level, (**D**) GPX4 activity, and (**E**) GPX4 protein expression. Treatment with cathelicidin significantly reversed the hyperoxia-induced changes in the ferroptosis biomarkers. *n* = 6–8 rats in each group. * *p* < 0.05, ** *p* < 0.01, and *** *p* < 0.001.

**Figure 4 antioxidants-11-02405-f004:**
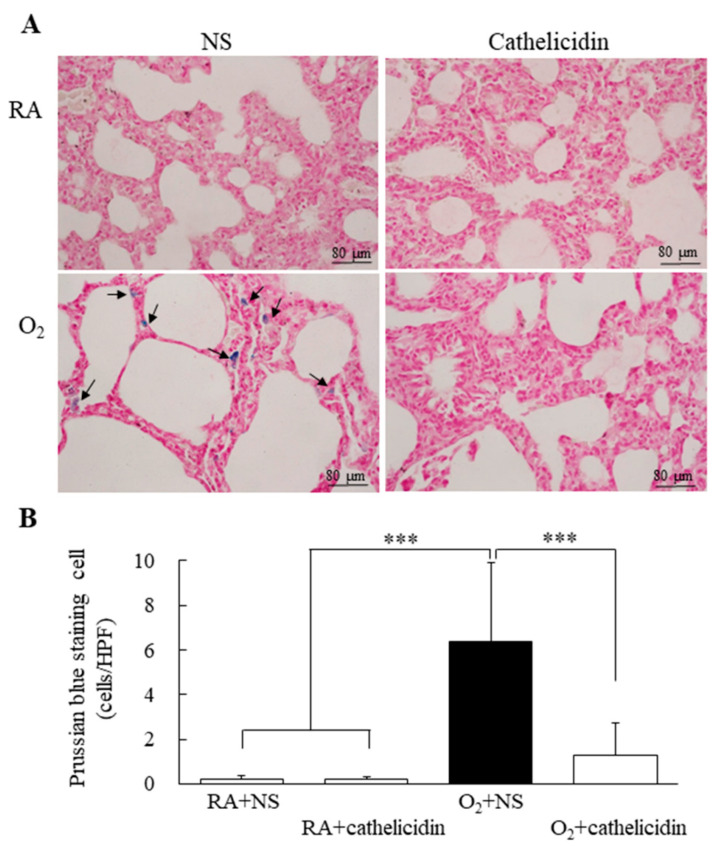
(**A**) Representative photomicrographs of Prussian blue staining and (**B**) positive cells per high-power field in 7-day-old rats exposed to RA or hyperoxia and treated with NS or cathelicidin. Prussian blue staining was primarily localized in type II alveolar cells and alveolar macrophages (black arrow). Cathelicidin treatment significantly reduced the hyperoxia-induced increase in the iron deposition. *n* = 8 rats in each group *** *p* < 0.001.

**Figure 5 antioxidants-11-02405-f005:**
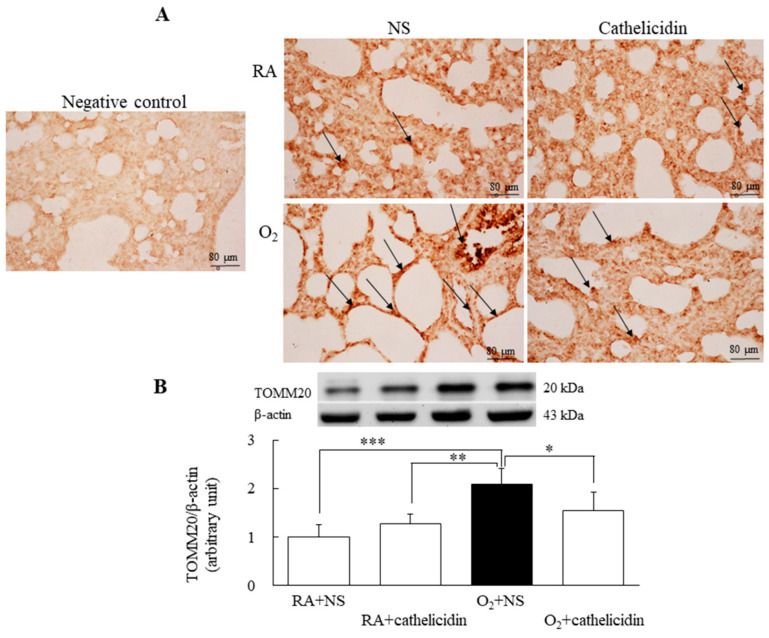
(**A**) Representative photomicrographs of immunohistochemistry of TOMM20 staining and (**B**) representative Western blots and quantitative data determined with densitometry for TOMM20 proteins in 7-day-old rats exposed to RA or hyperoxia and treated with NS or cathelicidin. The positive cells (black arrows) were present in the cytoplasm of the epithelial cell of bronchi, bronchioles, and alveoli, as well as vascular endothelial cells. Cathelicidin treatment significantly reduced TOMM20 expression in the hyperoxia-exposed group. *n* = 6 rats in each group. * *p* < 0.05, ** *p* < 0.01, and *** *p* < 0.001.

**Figure 6 antioxidants-11-02405-f006:**
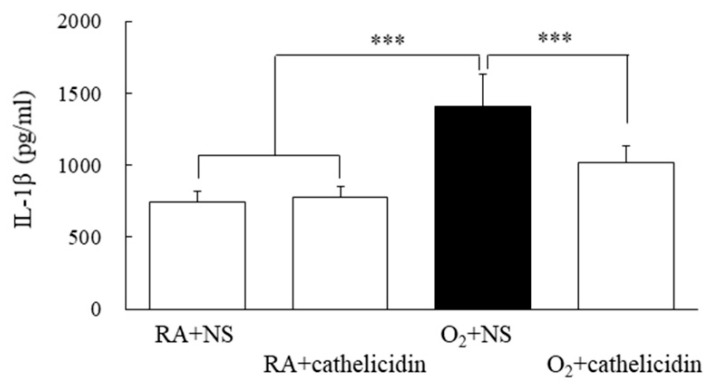
Lung IL-1β levels in 7-day-old rats exposed to RA or hyperoxia and treated with NS or cathelicidin. Treatment with cathelicidin significantly reduced the hyperoxia-induced increase in lung IL-1β levels. *n* = 6 rats in each group. *** *p* < 0.001.

**Figure 7 antioxidants-11-02405-f007:**
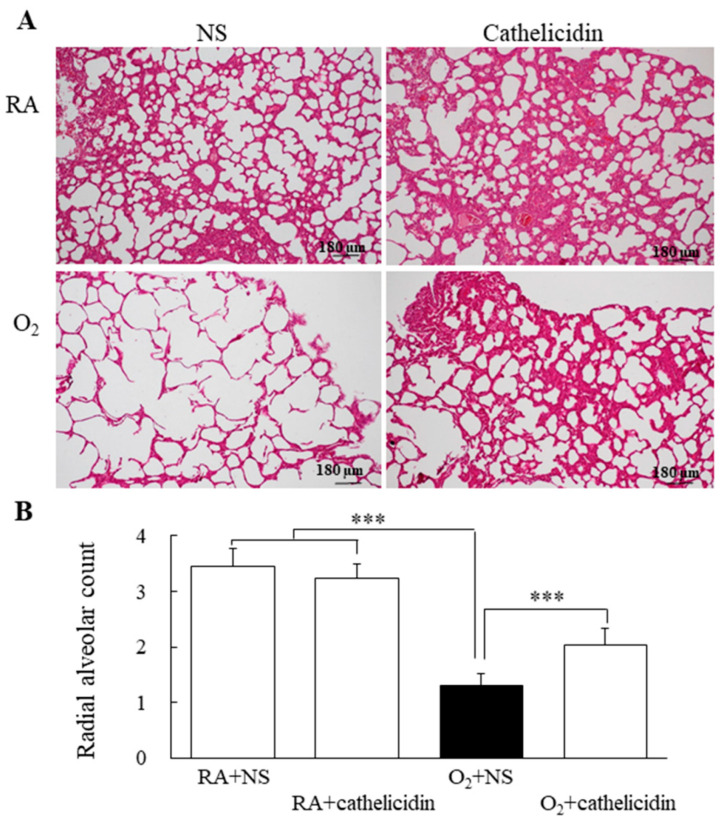
(**A**) Representative H&E-stained lung sections and (**B**) RAC in 7-day-old rats exposed to RA or hyperoxia and treated with NS or cathelicidin. The rats reared in RA and treated with NS or cathelicidin exhibited normal lung morphology and comparable RAC. Treatment with cathelicidin significantly reversed the hyperoxia-induced decrease in the RAC. *n* = 9–10 rats in each group. *** *p* < 0.001.

## Data Availability

All of the data is contained within the article and the Appendix A.

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
