# Peer review of "Cathelicidin Attenuates Hyperoxia-Induced Lung Injury by Inhibiting Ferroptosis in Newborn Rats"

_antioxidants, 2022, doi:10.3390/antiox11122405_

Round 1

Reviewer 1 Report

The study was focused on investigating the effect of Cathelicidin on Hyperoxia-induced Lung Injury. Supraphysiological oxygen is often required to treat newborns with respiratory disorders however, hyperoxia exposure during the first 7 days after birth can result in development of lung injury, decrease of body weights, development of ferroptosis, impaired lung alveolarization, iron deposition in the lungs and finally increased lung cytokine.

The authors introduced in vivo model of newborn rats which were reared in either room air (RA) or hyperoxia (85% O2) and then treated with cathelicidin (8 mg/kg) or normal saline (NS). Authors measured several factors of hyperoxia-induced lung injury such as iron deposition, presence of marker of lipid peroxidation - malondialdehyde, mitochondrial damage (TOMM20), concentration of interleukin-1β (IL-1β) in the lung area (lung homogenates); amount of glutathione or glutathione peroxidase 4, and radial alveolar count (RAC).

The structure of the paper is correct, and conclusions are clear and supported by well-set experiments. However, I have some points that need to be explained:

1/ what was the number of animals per each experimental group (please include in the Materials and Methods section and in each figure description)

2/ Figure 3 – Iron deposition / Prussian staining, please explain how the number of positive cells was calculated (how many fields was analyzed to get enough data to perform STAT analysis)

3/ Figure 4B – figure contains Western blotting of TOMM20 and figure TOMMS20 arbitrary units. Please explain what kind of data is included in the figure 4B, is this a densitometry calculated based on the Western Blot?

4/ Figure 6 – how was RAC calculated (number of pictures, magnification) to get enough data to perform STAT

Author Response

Author Response to Reviewer 1 Comments:

The structure of the paper is correct, and conclusions are clear and supported by well-set experiments. However, I have some points that need to be explained:

Response: Thanks for the reviewer’s comments.

1/ what was the number of animals per each experimental group (please include in the Materials and Methods section and in each figure description)

Response: Thanks for the reviewer’s comments. We added “Four female rats gave birth to 40 pups. Within 12 h of birth, the litters were pooled, and 10 pups were randomly redistributed among the new mothers” and the number of animals per each experimental group in Materials and Methods section and in Figure legends of revised version.

2/ Figure 3 – Iron deposition / Prussian staining, please explain how the number of positive cells was calculated (how many fields was analyzed to get enough data to perform STAT analysis)

Response: Iron deposition was quantified by counting the positive Prussian blue staining cells in randomly selected 50 high power fields (magnification 400´)/group, which were captured and transferred to a computer screen for analysis using the Image Pro Plus software (Media Cybernetics, Silver Spring, USA). Data generated from all sampled sections, fields, and counts were used for final calculation. We added these in 2.3. Determination of Iron Deposition section.

3/ Figure 4B – figure contains Western blotting of TOMM20 and figure TOMMS20 arbitrary units. Please explain what kind of data is included in the figure 4B, is this a densitometry calculated based on the Western Blot?

Response: We added “Densitometric analysis was performed to measure the intensity of TOMM20 and β-actin bands on AIDA. The densitometry unit of the TOMM20 protein expression in the RA + NS group was assigned as 1 after normalizing with β-actin” and “representative western blots and quantitative data determined with densitometry for TOMM20 proteins” in 2.4. Western Blots of GPX4 and TOMM20 and Figure legend 4B, respectively.

4/ Figure 6 – how was RAC calculated (number of pictures, magnification) to get enough data to perform STAT.

Response: RAC was measured by counting the number of alveoli passed by a vertical line from the center of the respiratory bronchioles to the border of the pleura (Figure S1). We identified one RAC measure as one unit and detected 90–100 units (magnification 200´)/group. Data generated from all sampled sections, fields, and counts were used for final calculations. We added these in 2.6. Lung Histology section and Supplementary Figure 1.

Reviewer 2 Report

In the present study, Chou et al. evaluated the protective effects of cathelicidin treatment on hypoxia-induced lung injury in newborn rats.  Rat pups were either exposed to room air (RA) or hyperoxia (85% O2) and then either treated with cathelicidin in saline (NS) or NS starting from postnatal days 1-6 in 4 groups: RA+NS, RA+ cathelicidin, O2+ NS, and O2+ cathelicidin. On a postnatal day 7, lungs were harvested for molecular and biochemical testing. Control and hyperoxia groups exhibited significantly lower body weight; higher Fe2+, iron deposition, mitochondrial damage marker TOMM20, and inflammatory marker- IL-1β and significantly lower glutathione, glutathione peroxidase 4, and radial alveolar count (RAC) when compared with the RA+NS or RA+cathelicidin. Cathelicidin treatment attenuated hyperoxia-induced lung injury, as indicated by higher RAC and lower TOMM20 and IL-1β levels and decreased ferroptosis. These findings indicated that cathelicidin attenuated hyperoxia-induced lung injury by inhibiting ferroptosis.

Though the experiments clearly demonstrate the observations, the major drawback is that the story lacks the in-depth mechanism and treatment specificity at the cellular and molecular levels. I have the following suggestions and comments to the authors to improve the current manuscript.

Major:

1.      The authors have earlier published the oxidative stress pathway for the same conditions, lung phenotypes, and protective effects of treatment in lung tissues. Cathelicidin is expressed by epithelial cells and neutrophils, so it would be important to check the indigenous expression of it in lungs tissue sections upon hyperoxia alone treatment which could be confirmed by co-immunostaining of major lung cell types and cathelicidin in control and hyperoxia conditions to see which cell types are being targeted and would benefit the most by the treatment.

2.      It would be important to provide more in-depth cellular level mechanisms as a bit of several cell type-related mechanisms would not help much to understand the treatment specificity of the protective effect. Authors have stated that alveolar cells and macrophages are in reference to ferroptosis and iron deposition, but the upstream or downstream signaling mechanism in these cells is not clear. Is mitochondrial damage also seen more in alveolar cells than in other cells?

3.      In the current presentation, several parallel mechanisms are being proposed, e.g., oxidative stress, mitochondrial injury, cytokine production, iron deposition, etc., some of which are known in other cells or organs. It would be better to focus on one or two pathways to explore the full signaling cascade rather than showing several parallel mechanisms leading to protective effects. For instance, lung cytokine IL1b levels are affected by cathelicidin treatment, which can be seen in the light of its known TLR-9-mediated effects in other organs, e.g., the intestine. To complete the story, from receptors to the signaling molecules to the cell function/ organ phenotype, it would be nice first to know the interconnection whether the cells expressing high IL1B levels are alveolar cells where the ferroptosis took place or if high IL1b expression resulted due to enhanced immune cell infiltration into lungs upon hyperoxia exposure and then explore the target receptors of Cathlecidin in specific lung cells to mediate the protective effects.  

4.      Radial alveolar counts are partially reversed by this treatment; it is unclear if the treatment dose is still suboptimal to reverse the damage. To see the dose-dependent effect, it would be good to try a higher dose to see if that completely reverses the RAC counts compared to hyperoxia.

Minor:

1.      Page 7, line 220: Treatment with cathelicidin significantly increased the hyperoxia-induced decrease in RAC should be written as Treatment with cathelicidin reversed the hyperoxia-induced decrease in RAC.

2.      Page 9, line 286 states: Currently, no effective therapy is clinically 286 available to prevent hyperoxia-induced kidney injury. Cathelicidin therapy may serve as 287, a novel strategy for preventing hyperoxia-induced lung injury. As the current manuscript is relevant to lung injury, do the authors intend to mention lung injury instead of kidney in the former sentence?

3.      Please include the negative staining controls with the immunostaining rep. images and add the scale bar to all images.

Author Response

Author Response to Reviewer 2 Comments:

Cathelicidin treatment attenuated hyperoxia-induced lung injury, as indicated by higher RAC and lower TOMM20 and IL-1β levels and decreased ferroptosis. These findings indicated that cathelicidin attenuated hyperoxia-induced lung injury by inhibiting ferroptosis.

Response: Thanks for the reviewer’s comments.

Though the experiments clearly demonstrate the observations, the major drawback is that the story lacks the in-depth mechanism and treatment specificity at the cellular and molecular levels. I have the following suggestions and comments to the authors to improve the current manuscript.

Major:

  1. The authors have earlier published the oxidative stress pathway for the same conditions, lung phenotypes, and protective effects of treatment in lung tissues. Cathelicidin is expressed by epithelial cells and neutrophils, so it would be important to check the indigenous expression of it in lungs tissue sections upon hyperoxia alone treatment which could be confirmed by co-immunostaining of major lung cell types and cathelicidin in control and hyperoxia conditions to see which cell types are being targeted and would benefit the most by the treatment.

Response: We have performed immunohistochemical staining for cathelicidin in lung tissue sections and added these in the Abstract, Methods, Results, and Figure 2 of the revised manuscript.

  1. It would be important to provide more in-depth cellular level mechanisms as a bit of several cell type-related mechanisms would not help much to understand the treatment specificity of the protective effect. Authors have stated that alveolar cells and macrophages are in reference to ferroptosis and iron deposition, but the upstream or downstream signaling mechanism in these cells is not clear. Is mitochondrial damage also seen more in alveolar cells than in other cells?

Response: Thanks for the reviewer’s comments. We added “It is less clear whether hyperoxia alters mitochondria in cells. Neonatal mice subjected to hyperoxia exhibit mitochondrial injury with complex-I malfunction, reduced ATP levels, and cell death in the lungs, according to Ratner et al [29]. Kandasamy et al. found that hyperoxia-induced lung development stop in newborn mice was associated with a decrease in mitochondrial activity and mitochondrial DNA variation [30]. Because cell reproduction and differentiation require energy, mitochondrial failure is most likely to blame for lung growth stop, which affects both the alveolar and vascular compartments” in Discussion section of the revised version.

  1. Ratner, V.; Starkov, A.; Matsiukevich, D.; Polin, R. A.; Ten. V. S. Mitochondrial dysfunction contributes to alveolar developmental arrest in hyperoxia-exposed mice. Am. J. Respir. Cell Mol. Biol. 2009, 40, 511-518.
  2. Kandasamy, J.; Rezonzew, G.; Jilling, T.; Ballinger, S.; Ambalavanan, N. Mitochondrial DNA variation modulates alveolar development in newborn mice exposed to hyperoxia. Am. J. Physiol. Lung Cell Mol. Physiol. 2019, 317, L740-L747.

  1. In the current presentation, several parallel mechanisms are being proposed, e.g., oxidative stress, mitochondrial injury, cytokine production, iron deposition, etc., some of which are known in other cells or organs. It would be better to focus on one or two pathways to explore the full signaling cascade rather than showing several parallel mechanisms leading to protective effects. For instance, lung cytokine IL1b levels are affected by cathelicidin treatment, which can be seen in the light of its known TLR-9-mediated effects in other organs, e.g., the intestine. To complete the story, from receptors to the signaling molecules to the cell function/ organ phenotype, it would be nice first to know the interconnection whether the cells expressing high IL1B levels are alveolar cells where the ferroptosis took place or if high IL1b expression resulted due to enhanced immune cell infiltration into lungs upon hyperoxia exposure and then explore the target receptors of Cathlecidin in specific lung cells to mediate the protective effects.

Response: Thanks for the reviewer’s comments. We added these as limitations “This research has some limitations. First, the cellular origin of IL-1êžµ expression in the lung tissues of newborn rats exposed to hyperoxia was not investigated though Piedboeuf et al. found no IL-1êžµ expression in the lungs of adult male mice exposed to room air. After 3 and 4 days of hyperoxia exposure, IL-1êžµ transcripts were found in pulmonary interstitial macrophages and neutrophils and were widespread in lung tissues [31]. Second, we did not concentrate on one or two pathways and investigate the entire signaling cascade rather than demonstrating several parallel mechanisms leading to protective effects” in the Discussion section of the revised version.

  1. Piedboeuf, B.; Horowitz. S.; Johnston, C. J.; Gamache, M.; Bélanger, S.; Poubelle, P. E.; Welty, S. E.; Watkins, R. H. Interleukin-1 expression during hyperoxic lung injury in the mouse. Free Radic. Biol. Med. 1998, 24, 1446-1454.

  1. Radial alveolar counts are partially reversed by this treatment; it is unclear if the treatment dose is still suboptimal to reverse the damage. To see the dose-dependent effect, it would be good to try a higher dose to see if that completely reverses the RAC counts compared to hyperoxia.

Response: In our study, the rats reared in hyperoxia and treated with NS exhibited a significantly lower RAC (1.31 ± 0.20) than did those reared in RA and treated with NS (3.44 ± 0.34) or cathelicidin (3.24 ± 0.26). Treatment with cathelicidin (8 mg/kg, 2.04 ± 0.30) significantly reversed the hyperoxia-induced decrease in RAC (p< 0.001, 1.31 ± 0.20). We counted the RAC with lower dose (4 mg/kg) and found the RAC values in the six groups are (1) RA + NS:   3.44 ± 0.34; (2) RA + cathelicidin (8 mg/kg): 3.24 ± 0.26; (3) RA + cathelicidin (4 mg/kg): 3.31 ± 0.19; (4) O2 + NS: 1.31 ± 0.20; (5) O2 + cathelicidin (8 mg/kg): 2.04 ± 0.30; (6) O2 + cathelicidin (4 mg/kg): 1.66 ± 0.11. These results suggest a trend toward the dose-dependent effect of cathelicidin and RAC.

Minor:

  1. Page 7, line 220: Treatment with cathelicidin significantly increased the hyperoxia-induced decrease in RAC should be written as Treatment with cathelicidin reversed the hyperoxia-induced decrease in RAC.

Response: We have corrected the sentence to “Treatment with cathelicidin significantly reversed the hyperoxia-induced decrease in the RAC” on lines 256-257 in the revised version.

  1. Page 9, line 286 states: Currently, no effective therapy is clinically 286 available to prevent hyperoxia-induced kidney injury. Cathelicidin therapy may serve as 287, a novel strategy for preventing hyperoxia-induced lung injury. As the current manuscript is relevant to lung injury, do the authors intend to mention lung injury instead of kidney in the former sentence?

Response: Thanks for the reviewer pointing out the typo. We corrected the word in the revised version.

  1. Please include the negative staining controls with the immunostaining rep. images and add the scale bar to all images.

Response: We added negative staining controls in the revised Figure 5 and added the scale bar to all images in the revised Figures 2, 3, 4, and 5 and Supplementary Figure 1.

Reviewer 3 Report

The present study is based on the evaluation of the effect of cathelicidin in hyperoxia-induced lung injury. Similar studies are published (https://pubmed.ncbi.nlm.nih.gov/32057991/) that have demonstrated similar mechanisms. For the study conducted on animals, the approval number granted by the ethics and deontology commission is missing (line 64). The materials and methods do not specify the number of pregnant females included in the study, the number of newborns that were used for the study. This is mentioned only in the results section. What is the reason that the newborns were mixed after birth and then redistributed to the females? Do you not consider that this can lead to the appearance of maternal aggression? For the assessment of alterations induced by hypoxia, do you consider the TOMM20 marker to be sufficient? Iron Deposition occurs only at lung level or can it be detected in other organs as well?Besides IL1 beta, wouldn't it have been necessary to perform the evaluation for IL-6, interleukin-10?

Author Response

Author Response to Reviewer 3 Comments:

The present study is based on the evaluation of the effect of cathelicidin in hyperoxia-induced lung injury. Similar studies are published (https://pubmed.ncbi.nlm.nih.gov/32057991/) that have demonstrated similar mechanisms.

Response: Thanks for the reviewer’s comments.

For the study conducted on animals, the approval number granted by the ethics and deontology commission is missing (line 64).

Response: The approval number (LAC-2018-0342) was added on lines 63-64.

The materials and methods do not specify the number of pregnant females included in the study, and the number of newborns that were used for the study. This is mentioned only in the results section.

Response: We added “Four female rats gave birth to 40 pups. Within 12 h of birth, the litters were pooled, and 10 pups were randomly redistributed among the new mothers” on lines 67-68 in 2. Materials and Methods: 2.1. Experimental Groups and the number of animals per experimental group in Figure legends of the revised version.

What is the reason that the newborns were mixed after birth and then redistributed to the females? Do you not consider that this can lead to the appearance of maternal aggression?

Response: Thanks for the reviewer’s suggestions. We pooled and randomized the rat pups to eliminate litter differences and bias and to equalize the number of runts in each group. There were no significant differences in mean maternal weights or litter sizes before pooling.

For the assessment of alterations induced by hypoxia, do you consider the TOMM20 marker to be sufficient?

Response: Thanks for the reviewer’s comments. We added this as a limitation “Third, other markers for mitochondrial dysfunction such as oxidative stress and impaired mitochondrial biogenesis were not measured. Though oxidative stress and TOMM20 have been reported to be implicated in hyperoxia-induced lung injury [15,28]” in the Discussion section of the revised version.

Iron Deposition occurs only at lung level or can it be detected in other organs as well?

Response: The effects of hyperoxia on iron deposition in other organs were mostly unknown. We added “The consequence of hyperoxia on iron deposition in other organs was mostly unknown. Ferroptosis has been linked to a variety of system diseases, including nervous system diseases, heart disease, liver disease, gastrointestinal disease, lung disease, kidney disease, and others [23]. Almost all organs and tissues have been shown to be targets of oxygen toxicity, though some have received less attention [24]. Iron deposition was discovered in other organs in theory. More research is needed to investigate the effects of hyperoxia on iron deposition in other organs.” in the Discussion section of the revised version.

  1. Li, J.; Cao, F.; Yin, H. L.; Huang, Z. J.; Lin, Z. T.; Mao, N.; Sun, B.; Wang, G. Ferroptosis: past, present and future. Cell Death Dis. 2020, 11, 88.
  2. Alva, R.; Mirza, M.; Baiton, A.; Lazuran, L.; Samokysh, L.; Bobinski, A.; Cowan, C.; Jaimon, A; Obioru, D.; Al Makhoul, T.; et al. Oxygen toxicity: cellular mechanisms in normobaric hyperoxia. Cell Biol. Toxicol. 2022, 16, 1–33.

Besides IL1 beta, wouldn't it have been necessary to perform the evaluation for IL-6, interleukin-10?

Response: Thanks for the reviewer’s suggestions. As IL-10 is one of the anti-inflammatory cytokines (1), and the modulation of lymphocytic response like B-cell proliferation is strictly seen as an effect of IL-6 that is induced by IL-1β in an animal model with either IL-6 or IL-1β knockout mice (2). Recent studies demonstrate that IL-1β contributes to lung injury in a wide range of relevant studies (3-5). The aim of this study was to use a representative marker of the hyperoxia-induced inflammatory response, and we chose to measure IL-1β for the reasons described above. 

  1. Banchereau, J.; Pascual, V.; O’Garra, A. From IL-2 to IL-37: The expanding spectrum of anti-inflammatory cytokines. Nat. Immunol. 2012, 13, 925–931.
  2. Rosser, E. C.; Oleinika, K.; Tonon, S.; Doyle, R.; Bosma, A.; Carter, N. A.; Harris, K. A.; Jones, S. A.; Klein, N.; Mauri, C. Regulatory B cells are induced by gut microbiota-driven interleukin-1 and interleukin-6 production. Nat. Med. 2014, 20, 1334–1339.
  3. Li, D.; Ren, W.; Jiang, Z.; Zhu, L. Regulation of the NLRP3 inflammasome and macrophage pyroptosis by the p38 MAPK signaling pathway in a mouse model of acute lung injury. Mol. Med. Rep. 2018, 18, 4399–4409.
  4. Mahmutovic Persson, I.; Menzel, M.; Ramu, S.; Cerps, S.; Akbarshahi, H.; Uller, L. IL-1β mediates lung neutrophilia and IL-33 expression in a mouse model of viral-induced asthma exacerbation. Respir. Res. 2018, 19, 16.
  5. Liu, Q.; Tian, X.; Maruyama, D.; Arjomandi, M.; Prakash, A. Lung immune tone via gut-lung axis: gut-derived LPS and short-chain fatty acids' immunometabolic regulation of lung IL-1β, FFAR2, and FFAR3 expression. Am. J. Physiol. Lung Cell Mol. Physiol. 2021, 321, L65–L78.

Round 2

Reviewer 3 Report

Most of my observations have been corrected and supplemented accordingly.